# Machine Learning-Based Prediction of Orphan Genes and Analysis of Different Hybrid Features of Monocot and Eudicot Plants

**Qijuan Gao [1], Xiaodan Zhang [2], Hanwei Yan [3] and Xiu Jin [3,\*]**

[1]   School of Computer Science, Hefei Normal University, Hefei 230001, China
[2]   Anhui Province Key Laboratory of Smart Agricultural Technology and Equipment, Anhui Agricultural University, Hefei 230036, China
[3]   Key Laboratory of Crop Biology of Anhui Province, Anhui Agricultural University, Hefei 230036, China
\*   Correspondence: jinxiu123@ahau.edu.cn

**Abstract:** Orphan genes (OGs) may evolve from noncoding sequences or be derived from older coding material. Some shares of OGs are present in all sequenced genomes, participating in the biochemical and physiological pathways of many species, while many of them may be associated with the response to environmental stresses and species-specific traits or regulatory patterns. However, identifying OGs is a laborious and time-consuming task. This paper presents an automated predictor, XGBoost-A2OGs (identification of OGs for angiosperm based on XGBoost), used to identify OGs for seven angiosperm species based on hybrid features and XGBoost. The precision and accuracy of the proposed model based on fivefold cross-validation and independent testing reached 0.90 and 0.91, respectively, outperforming other classifiers in cross-species validation via other models, namely, Random Forest, AdaBoost, GBDT, and SVM. Furthermore, by analyzing and subdividing the hybrid features into five sets, it was proven that different hybrid feature sets influenced the prediction performance of OGs involving eudicot and monocot groups. Finally, testing of small-scale empirical datasets of each species separately based on optimal hybrid features revealed that the proposed model performed better for eudicot groups than for monocot groups.

**Keywords:** orphan genes (OGs); hybrid features; machine learning; angiosperm



## 1. Introduction

Monocotyledonous and eudicotyledonous plants (monocots and eudicots) have morphological differences in the number and arrangement of their embryonic leaves. These are typically parallel in monocots and reticulate in eudicots; besides, monocots have a sheathing leaf base encircling the stem. Monocots diverged from their eudicot relatives in angiosperm evolution derived from the whole genome duplication (WGD), which contributed to increased diversification, environmental adaptation, and genomic novelty [1]. In the evolutionary process, orphan genes (OGs) can arise in a lineage and are prevalently expressed in many organisms [2]. In particular, taxonomically restricted OGs are widely distributed in angiosperm species, including eudicot and monocot groups, such as *Arabidopsis thaliana, Populus trichocarpa, Citrus sinensis, Triticum aestivum, Oryza sativa, cowpea, Camellia sinensis,* and *Saccharum spontaneum* [3–10]. Numerous studies of OGs have identified general trends in the sequence features of OGs across different species, including gene length, GC content, and introns, which are also vital for environmental adaptation, including biotic and abiotic stress [11–13]. Specifically, the OG Qua-Quine Starch (QQS) in *Arabidopsis thaliana* is known to regulate the ratio of protein and starch carbon. Being transferred and expressed in other species, QQS has been reported to change the metabolic process by regulating the allocation of carbon and nitrogen in proteins and carbohydrates and affecting the compounds in seeds and leaves, consequently improving crop yields [14].

Previous studies also revealed that OGs play a vital role in response to drought stress in *cowpea* and *Fusarium resistance* in *Triticum aestivum* [6,7].

OGs have usually been applied through BLAST (Basic Local Alignment Search Tool) sequence alignment, involving genome and transcriptome sequences for all analysis processes, including BLASTP, BLASTN, TBLASTX, and so on [15]. However, this method is time-consuming and requires considerable server-driven resources to identify OGs. Alternatively, OGs can be distinguished from nonorphan genes (NOGs), e.g., protein-coding genes, by more significant differences in gene length, exon number, GC content, and expression level [11]. Their analysis and further classification can be facilitated via machine learning-based methods, which have already been successfully applied to classifying biological datasets and solving various discrimination problems. Thus, such ensemble learning methods as Gradient Boosting Decision Tree (GBDT), Random Forest, and Adaboost have been used for biological prediction based on genome datasets. In particular, Zhu et al. used GBDT to classify tissue and cell types in cancer samples using a gene expression dataset, which performed similarly to other machine learning methods [16]. In contrast, the Extreme Gradient Boost (XGBoost) method adopted by Chen and Guestrin [17] outperformed numerous machine learning methods and found wide applications in data mining, regression, and classification domains. In addition, Gao et al. have used an effective model named SMOTE-ENN-XGBoost to predict the OGs of *A. thaliana* [18]. However, to the best of the authors' knowledge, it has yet to be carried out in the bioinformatic field of predicting OGs of different types of plant species.

In this study, OGs were measured by taking into account sequence features, which share some characteristics of other angiosperm species (shorter sequence length, fewer exon numbers, and lower GC content), while having fewer transcript support and lower expression than NOGs [12]. Then, these protein features were extracted, and the XGBoost-A2OG model was constructed and applied to the prediction of OGs in angiosperm species.

## 2. Related Works

Recently, machine learning methods have received considerable interest in the identification of OGs fields, which are an important source of genetics and contribute to evolutionary innovations. These methods include the Decision tree (DT) [19], Neural network (NN) [19], Convolutional Neural Network (CNN) with transformer [20], and ensemble learning method [20]. Besides, many researchers have been conducted to compare different machine learning algorithms or combined with other methods to accelerate the performance of identification of OGs.

Gao et al. proposed a novel ensemble method to predict the OGs of A. thaliana in bioinformatics studies. Then another deep learning method, CNN with transformer technique was successfully applied to identifying OGs in moso bamboo which used a convolutional neural network in combination with a transform neural network in protein sequences [19]. Their proposed approach provides better performance in a specific species.

In addition, decision trees and neural networks were employed to improve the accurate discovery of OGs by Casola et al. relying on basic sequence features obtained from DNA and protein sequences in three angiosperm families. The experimental results showed that both DT and NN classifiers achieve high levels of accuracy and recall in identifying OGs.

Recently, many studies have confirmed that OGs generated de novo in a species may be more prevalent than gene duplication and be one of the main ways of orphan generation [21–25]. Some researchers have found that in the newly evolved OGs in Arabidopsis, protein length is usually shorter, mainly due to the evolution of the orphan gene having fewer exons in the process, while in some species, the exon length is significantly shorter [26,27].

However, these researchers haven't focused on different families of angiosperm plants. To find a general method to identify a large number of plants of OGs based on a rapid accumulation of genomic data, we have analyzed some features regarding the genome and protein sequences that may affect the results in the classification process.

## 3. Materials and Methods

### 3.1. The Framework of the XGBoost-A2OG Model

The workflow used in this study and depicted in Figure 1 comprised the following five parts: data selection, data pre-processing, data modelling, model development, and model interpretation.

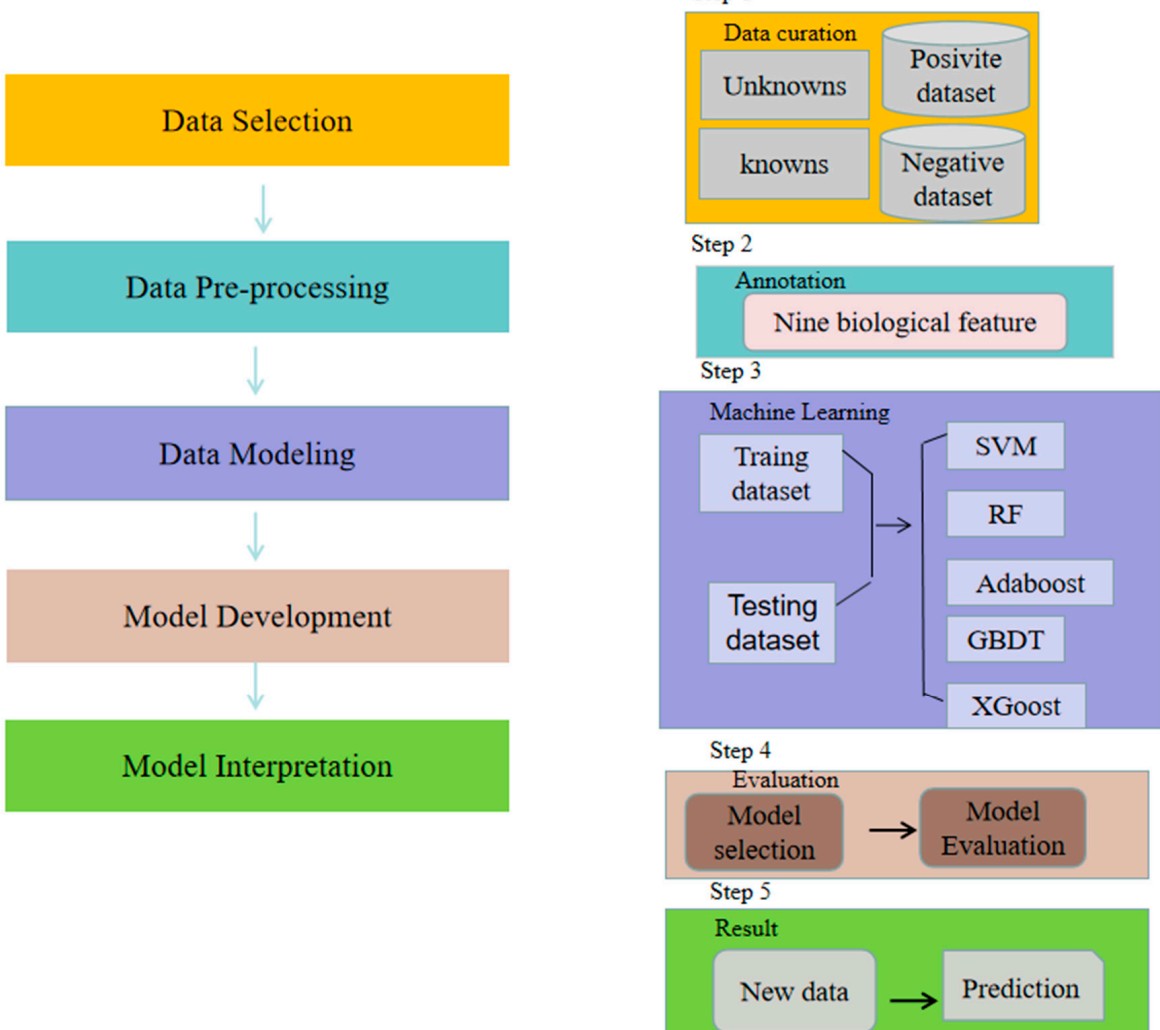

**Figure 1.** Workflow of the model framework.

### 3.2. Data Collection

This study collected protein sequences and gene annotation datasets for 136 plant species from Phytozome [28]. Non-redundant protein sequences (NR) were obtained from the NCBI database [29] and Ensemble Plants [30]. Next, BLASTp was used to identify OG based on a previous study [28] to search for homologs of all 401,834 gene annotations in seven plants (*Arabidopsis thaliana, Populus trichocarpa, Citrus sinensis, Camellia sinensis Sorghum bicolour, Oryza sativa, Zea mays*) (Figure 2) of the other 94 species released in Phytozome V12.1 with an E-value cutoff of $1 \times 10^{-3}$. Noteworthy is that the E-value or expectation value is a more inclusive value than probability, defining the number of times the query sequence is expected to match with the database sequences by random chance.

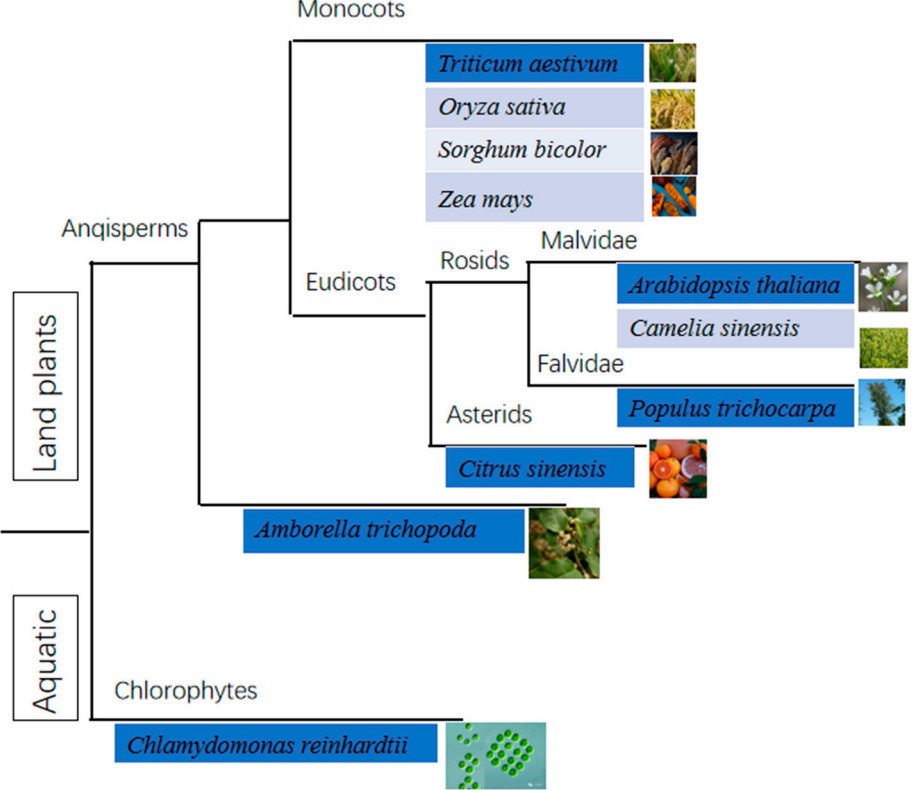

**Figure 2.** The phylogenetic tree of Monocotyledonous and eudicotyledonous plants.

The obtained 9022 OGs and 392,812 NOGs were identified with label 1 and label 0, respectively, to thoroughly train the ensemble learning model. All of them were combined to form the five plant species' OG datasets. Then, we extracted the characteristics of gene structure, cDNA sequence, and protein-coding genes of all five species from Phytozome and Ensemble plants, forming databases containing high annotation of plant genomes.

### 3.3. Ensemble Algorithm

XGBoost (Extreme Gradient Boost) is an ensemble learning technique for regression and classification problems based on the boosting algorithm [17]. The motivation is to classify data using the best hyperplane representing the most extensive separation between two classes. Unlike the traditional integrated decision tree algorithm, XGBoost adds a regular term to the loss function, which can control the complexity of the model and prevent the model from overfitting. The objective function is given to be optimized by the following formula:

(1) Taylor's formula to approximate the original goal.

$$obj(\theta) = \sum_{i=1}^{n} l(y_i, y_{i,}) \tag{1}$$

(2) Taylor expansion:

$$obj^{(t)} = \sum_{i=1}^{n} l(y_i, \hat{y}_i^{t-1}) + g_i f_t(x_i) + \frac{1}{2} h_i f_t^2(x_i)] + \Omega(f_t) + \text{constant} \cdots \tag{2}$$

(3) Among them, $g_i$, $h_i$ are expressed as:

$$\left\{ \begin{array}{l} g_i = \partial_{\hat{y}^{(t-1)}} l(y_i, \hat{y}^{(t-1)}), \\ h_i = \partial 2_{\hat{y}^{(t-1)}} l(y_i, \hat{y}^{(t-1)}). \end{array} \right\} \tag{3}$$

(4) The formula of decision tree complexity calculation:

$$\Omega(f) = \gamma T + \frac{1}{2}\lambda\sum_{j=1}^{T} w_j^2 \tag{4}$$

(5) $T$ is the number of leaf nodes, and $w$ is the leaf node score. Substituting (2)–(4) into (1) the objective function:

$$obj^{(t)} = \sum_{j=1}^{T} \left[ G_j w_j + \frac{1}{2}(H_j + \lambda)w_j^2 \right] + \gamma T \tag{5}$$

(6) Among, $I_{ij} = \{i|q(x_i = j)\}$, which represents the sample set belonging to the *j*-th leaf node.

$$G_j = \sum_{i \in I_j} g_i, H_j = \sum_{i \in I_j} h_i \tag{6}$$

(7) To minimize the objective function, set the derivative being 0 and find the optimal prediction score of each leaf node:

$$w_j^* = -\frac{G_j}{H_j + \lambda} \tag{7}$$

(8) Substitute the objective function again to get its minimum value:

$$obj^* = -\frac{1}{2}\sum_{j=1}^{T} \frac{G_j^2}{H_j + \lambda} + \gamma T \tag{8}$$

(9) Use *obj*\* to find the tree with the best structure and add it to the model and apply the greedy algorithm to find the optimal tree structure. Each time when trying to add a split to an existing leaf, the *Gain* is calculated as follows:

$$Gain(\Phi) = \frac{1}{2}\left[ \frac{\left(\sum\limits_{i \subseteq I_L} g_i\right)^2}{\sum\limits_{i \subseteq I_L} h_i + \lambda} + \frac{\left(\sum\limits_{i \subseteq I_R} g_i\right)^2}{\sum\limits_{i \subseteq I_R} h_i + \lambda} - \frac{\left(\sum\limits_{i \subseteq I} g_i\right)^2}{\sum\limits_{i \subseteq I} h_i + \lambda} \right] - \gamma \tag{9}$$

When the XGBoost model was used in the experiment, the following parameters were adjusted to make the model perform its best performance. For example, one of the most critical parameters in this and other tree-based ensemble algorithms, such as GBDT, Random Forest (RF), and AdaBoost, is "learning_rate", which dramatically affects the model performance. Another parameter is "n_estimators", which is the number of iterations in training: too small or too large parameters will lead to underfitting or overfitting, respectively. The third critical parameter is "max_depth", which is the maximum depth of the tree. Its higher values make the tree model more complex and improve its fitting ability, but at the same time, it increases the risk of overfitting.

In contrast to XGBoost, the GBDT is a radial basis function kernel that adopts an automatic gamma value (which is the inner product coefficient in the polynomial) and soft margin parameter $C = 1$, which controls the trade-off between the slack variable penalty and the margin size. Random Forest (RF) is based on trees and is characterized by the square root of the number of features. In AdaBoost, the most critical parameters are "base_estimator", "n_estimators", and "learning_rate".

### 3.4. Data Preparation and Feature Selection Settings

Data pre-processing is the base step before mining data, including cleanup, integration, and transformation, as well as data discretization, missing value, and outlier processing. The first pre-processing stage focuses on detecting incomplete, accurate, consistent, and

corrupt data and then modifying or deleting these false data with some techniques. Different datasets have various characteristics in actual research, so there are different ways to predict the data.

In this paper, we divide into two parts feature selection, one is the filter-based feature selection. This algorithm adopts some principles involving information, consistency, dependency, and distance for measuring the feature characteristics, which are generalized for various classifiers based on the independent features of the machine learning algorithm [31]. For example, a variation filter is to remove the features with small difference value and retain the features with large variance value, because the variance of each feature determines the different degree of the feature in a sample. When a feature in the data set is exposed to Bernoulli distribution (binary classification), it can be used the formula as follow:

$$\sigma = p(1 - p) \tag{10}$$

The classic Chi-square(Chi2) filter method is a statistical test for computing the correlation from two types of categorical data. Considering the inconsistency between the observed value and the expected value of the sampling frequency, such as the independent variable equal to i and the dependent variable equal to j, the statistic is constructed, Chi2 tests use the following formula to calculate the test statistic:

$$\kappa^2 = \frac{(A - E)^2}{E} \tag{11}$$

The other part is manual feature selection. In this section, we set three main experiments to evaluate the classifiers to validate the performance of classifying the OGs from various feature datasets with the proposed model.

Firstly, two sets of experiments were organized based on nine gene pair feature datasets involving GC, GC%, protein length, molecular weight ($Mw$(Da)), isoelectric point value ($pI$), exon number, average exon length, intron number, average intron length, gene length, and the output value as an assessment criterion, namely, AGI, for detecting the conditional relatedness between a pair of genes. For model training, the datasets were divided into two sections containing training and testing parts, and the target labels of AGI values were marked as 1 s and 0 s for the two types of gene pairs. The total datasets were divided into training, validating and testing processes using 5-fold cross-validation. The training dataset was used to develop the aforementioned statistical criteria for selected models. The testing dataset was applied to assess the performance of these models with the default parameters without tuning.

Secondly, to explore the importance of genomic and cDNA sequence features after selecting the optimal models, we used a feature selection method by removing one feature from "set_all" of features each time with no redundancy, such as set1 of feature data with no protein length, set2 with no protein of $Mw$(Da), set3 with no protein of $pI$, set4 with no exon number, and set5 with no GC%.

Finally, to validate this model for predicting the OGs of each plant species with specific feature sets, we selected seven testing datasets matched with seven plants (*Arabidopsis thaliana*, *Populus trichocarpa*, *Sorghum bicolour*, *Oryza sativa*, *Zea mays*, *Citrus sinensis*, and *Camellia sinensis*).

### 3.5. Validation Strategies and Evaluation Metrics

The confusion matrix is a matrix table (shown in Table 1) that is used to judge the validation of classification. The results of the prediction model are analyzed using four basic indicators: true positive (TP), true negative (TN), false positive (FP), and false negative (FN).

**Table 1.** Binary confusion matrix.

|  | Real Positive | Real Negative |
|---|---|---|
| Predict positive | TP | FP |
| Predict negative | FN | TN |

We performed an initial statistical analysis to evaluate the prediction performance for binary classes and grasp the critical features. As the performance measures, stratified five-fold cross-validation was used for obtaining classification accuracy; however, accuracy was found to be an inappropriate evaluation metric for class-imbalanced datasets. Alternatively, precision, recall, F1-Score, and AUC (area under the ROC curve) parameters were used to evaluate the proposed method's feasibility, as in [32]. The AUC is the value of the area under the ROC curve (receiver operating characteristic) that reflects the probability of identifying correct and wrong results according to different thresholds, which is generally between 0.5–1. The quantized index value can better compare the performance of the classifiers: a high-performance classifier AUC value is close to 1properly reflected the test performance.

(i) Accuracy rate (accuracy rate of positive samples):

$$\text{Accuracy} = \frac{\text{TP} + \text{TN}}{\text{TP} + \text{TN} + \text{FP} + \text{FN}} \tag{12}$$

(ii) Recall rate (accuracy rate of positive samples):

$$\text{Recall} = \frac{\text{TP}}{\text{TP} + \text{FN}} \tag{13}$$

(iii) Precision (precision rate of positive samples):

$$\text{Precision} = \frac{\text{TP}}{\text{TP} + \text{FP}} \tag{14}$$

(iv) F1-score value:

$$\text{F1} = \frac{2\text{PR}}{\text{P} + \text{R}} \tag{15}$$

## 4. Results and Discussion

### 4.1. The Compared Features between OGs and NOGs in Different Species

After the above seven species of gene datasets were introduced, the next step was to arrange their variable features for constructing a prediction model. The annotations of all protein sequences, CD sequences, and gene characteristics included *GC*, *GC*%, gene length, *Mw* (Da), *pI*, average exon length, average intron length, and so on. Seven species were compared in regard to OGs and NOGs. As seen in Figure 3b–d, OGs had lower values of gene length, *Mw* (Da), and *GC*% than NOGs, with the opposite result on *pI* values (Figure 3a).

Another critical step was data selection. This paper selected nine features: (1) GC, (2) GC%, (3) protein length, (4) molecular weight (Mw (Da)), (5) isoelectric point value (pI), (6) exon number, (7) average exon length, (8) intron number, and (9) average intron length, which were denoted as 1 to 9, respectively, and recorded as V1–V9. The classes of orphan genes and nonorphan genes were recorded as 1 and 0, respectively. Since the datasets contained various types of features, and attribute units were inconsistent in dimensions, it was necessary to use the pre-processing method to standardize the data.

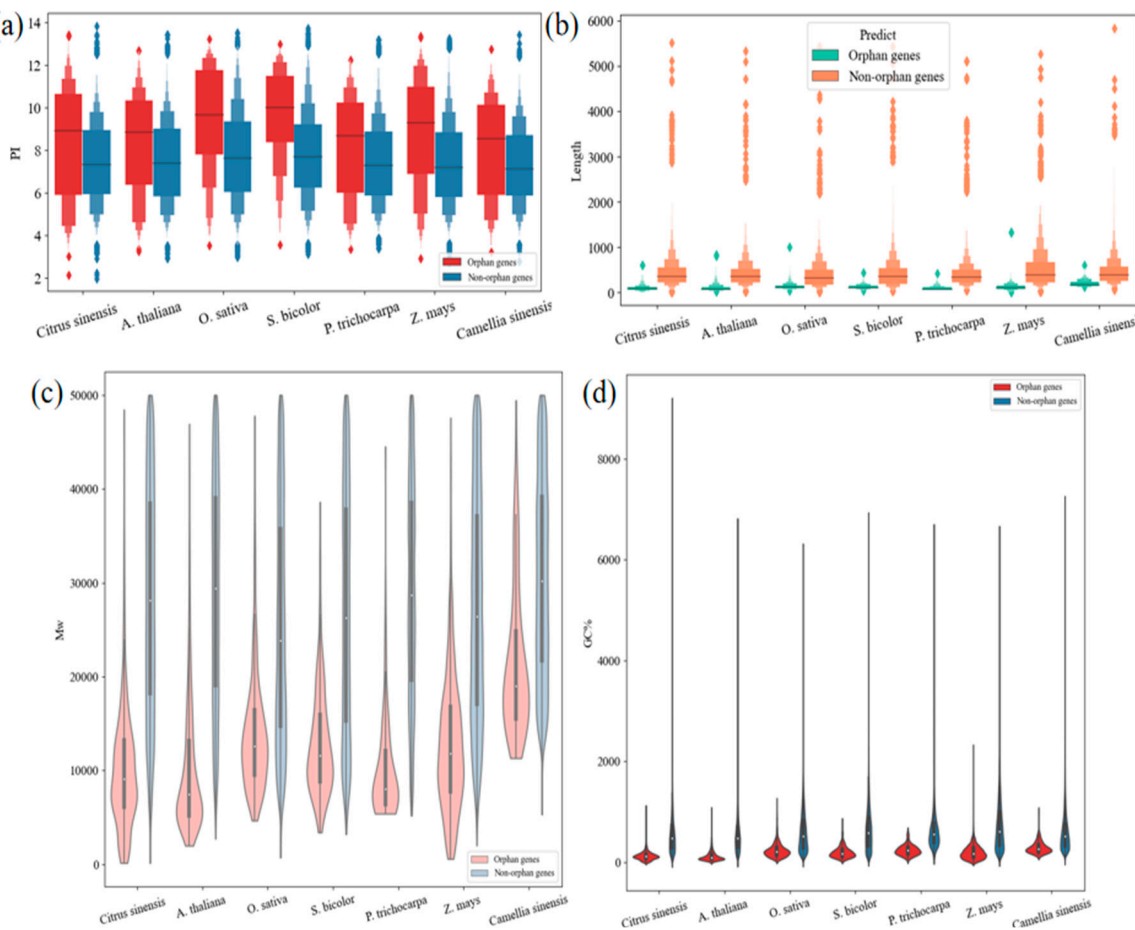

**Figure 3.** Comparison of various features of OGs and NOGs in seven species: (**a**) *pI*, (**b**) gene length, (**c**) *Mw*, (**d**) *GC%*.

### 4.2. Comparison with Other Methods for Predicting Cross-Species OGs

This study constructed a novel hybrid classification XGBoost-A2OG model for classifying the angiosperm species of OG distributions. We tested the dataset of *Arabidopsis thaliana*, *Populus trichocarpa*, *Sorghum bicolour*, *Oryza sativa*, *Zea mays*, *Citrus sinensis*, and *Camellia sinensis*, obtaining 6322 OGs from the public release of these species' protein sequences through the BLAST sequence alignment. To predict the OGs, XGBoost-A2OGs were trained using the following parameters: 200 estimators and a learning rate of 0.02 with a maximum depth of six. To optimize the parameters, the optimized XGBoost-A2OG models were trained by 5-fold nested cross-validation. In addition, we compared XGBoost-A2OGs with SVM and tree-based ensemble algorithms (GBDT, RF, and AdaBoost).

The results on the accuracy, precision, recall, and F1-score of the five models are listed in Table 2. Compared to the above four reference methods, the proposed XGBoost-A2OG model achieved competitive performance in recall and F1-score, outperforming them in precision. Thus, it more precisely distinguished normal OGs from NOGs, exhibiting the best classification effect on the AG datasets.

**Table 2.** Performance measure indices of the five models based on the same parameters of the training and test datasets.

| Index | SVM | RF | GBDT | AdaBoost | XGBoost |
|-------|-----|-----|------|----------|---------|
| Accuracy | 0.88 | 0.85 | 0.88 | 0.88 | 0.91 |
| Precision | 0.86 | 0.79 | 0.87 | 0.87 | 0.90 |
| Recall | 0.91 | 0.97 | 0.89 | 0.88 | 0.91 |
| F1-Score | 0.88 | 0.87 | 0.88 | 0.88 | 0.91 |

Moreover, according to the area under the curve (AUC) value shown in Figure 4, the ROC and precision-recall (P-R) curves of the XGBoost model completely wrapped those of the other four methods (AdaBoost, GBDT, RF, and SVM), outperforming them by classification efficiency.

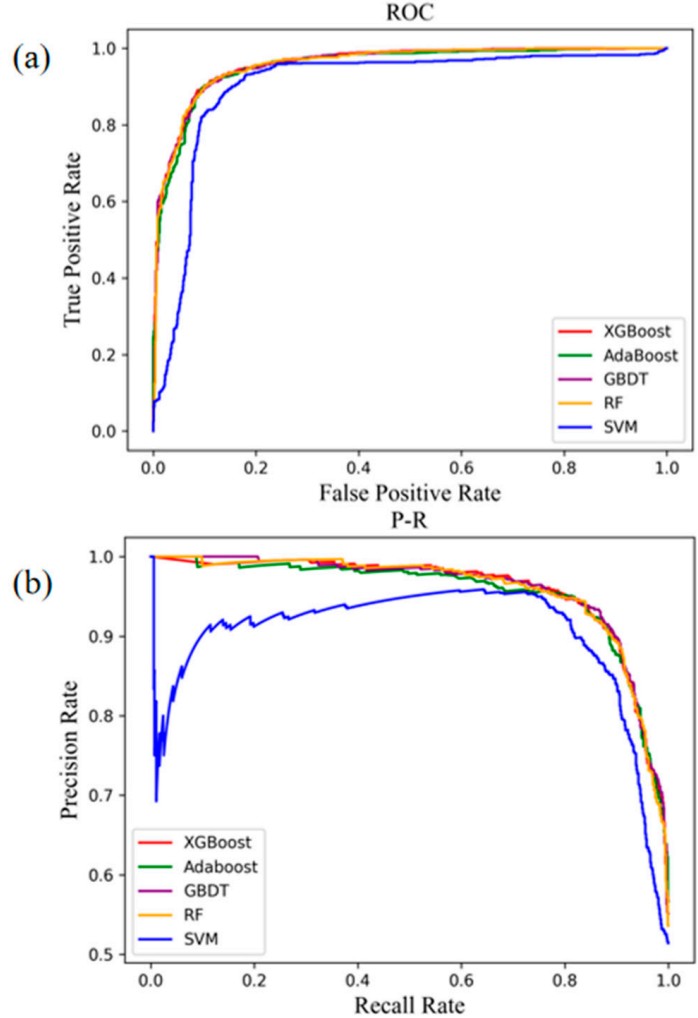

**Figure 4.** The ROC (**a**) and precision-recall (P-R) (**b**) curves of XGBoost, Adaboost, GBDT, RF, and SVM methods.

*4.3. Predicting OGs with Different Feature Sets in Eudicot and Monocot Species via XGBoost-A2OGs*

Some features might become noise, deteriorating the robustness and stability of the constructed model. Moreover, contribution rates of various features differ, the highest ones being the most lucrative for OGs' prediction. Therefore, this work presents two filter-based selection methods to remove irrelevant and redundant features in terms of both training processes. In particular, we selected two types of delegated species from the eudicot subclass (*P. trichocarpa* and *Camellia sinensis*) and monocot subclass (*O. sativa* and *S. bicolour*) applied with filter-based selection methods. Then the filtered feature are the same containing the GC, protein length, Mw (Da), and pI. Thus, the classification results on these selection methods with four species separately by variation and Chi2 method based on the XGBoost-A2OGs model are listed in Table 3.

**Table 3.** Performance measure indices of eudicot and monocot species for the training and testing datasets by filter method based on the same parameters.

| Type | Species | Filter Method | Precision | Accuracy | AUC |
|---|---|---|---|---|---|
| Eudicots | *P. trichocarpa* | variation | 0.92 | 0.93 | 0.94 |
| Eudicots | *P. trichocarpa* | Chi2 | 0.9 | 0.92 | 0.94 |
| Eudicots | *Camellia sinensis* | variation | 0.82 | 0.69 | 0.85 |
| Eudicots | *Camellia sinensis* | Chi2 | 0.82 | 0.69 | 0.85 |
| Monocots | *O. sativa* | variation | 0.78 | 0.83 | 0.9 |
| Monocots | *O. sativa* | Chi2 | 0.78 | 0.83 | 0.9 |
| Monocots | *S. bicolor* | variation | 0.81 | 0.87 | 0.94 |
| Monocots | *S. bicolor* | Chi2 | 0.81 | 0.87 | 0.94 |

Filter algorithms can scale for multiple dimensional datasets. However, the features selected by the filter method ignore the interaction among features, and individual scores in a filter-based method are assigned to each feature without considering its significance in combination with other shared features. Therefore, we further proposed an artificial group for feature selection to explore the contribution of each feature for different types of angiosperm. First of all, we also selected a eudicot subclass (*P. trichocarpa* and *Camellia sinensis*) and applied to them five sets of feature selection methods to identify the one with the optimal performance. The classification results on five sets of feature selection methods with two species separately based on XGBoost-A2OGs are listed in Table 4, where the Set3 of Camellia sinensis featured the lowest precision, accuracy, and AUC values (0.80, 0.69, and 0.85). Meanwhile, the Set5 of *P. trichocarpa* combined the highest respective values (precision of 0.9, accuracy of 0.92, and AUC = 0.94).

**Table 4.** Performance measure indices of eudicot species for the training and testing datasets by feature sets based on the same parameters.

| Type | Species | Feature | Precision | Accuracy | AUC |
|---|---|---|---|---|---|
| Eudicots | *P. trichocarpa* | Set_all | 0.9 | 0.9 | 0.92 |
| Eudicots | *P. trichocarpa* | Set1 | 0.89 | 0.87 | 0.89 |
| Eudicots | *P. trichocarpa* | Set2 | 0.9 | 0.9 | 0.91 |
| Eudicots | *P. trichocarpa* | Set3 | 0.88 | 0.9 | 0.92 |
| Eudicots | *P. trichocarpa* | Set4 | 0.9 | 0.9 | 0.94 |
| Eudicots | *P. trichocarpa* | Set5 | 0.9 | 0.92 | 0.94 |
| Eudicots | *Camellia sinensis* | Set_all | 0.89 | 0.74 | 0.85 |
| Eudicots | *Camellia sinensis* | Set1 | 0.83 | 0.68 | 0.84 |
| Eudicots | *Camellia sinensis* | Set2 | 0.83 | 0.69 | 0.82 |
| Eudicots | *Camellia sinensis* | Set3 | 0.80 | 0.69 | 0.85 |
| Eudicots | *Camellia sinensis* | Set4 | 0.94 | 0.76 | 0.87 |
| Eudicots | *Camellia sinensis* | Set5 | 0.89 | 0.74 | 0.88 |

As it was mentioned earlier, monocots have branched off from eudicots via whole genome duplication (WGD) [33]. Systematic identification of orphan genes in eudicots revealed that the optimal precision of *P. trichocarpa* and *Camellia sinensis* orphan genes were nearly 0.9 shown in Table 4. Five sets of feature selection methods were also applied to reveal the optimal feature selection performance with XGBoost-A2Ogs for the monocot group containing *O. sativa* and *S. bicolour*. The results are listed in Table 5, indicating that the Set5 feature selection in the monocot group yielded higher precision, accuracy, and AUC values than those obtained via the Set_all feature selection. The respective values of *S. bicolour* in Set5 (0.82, 0.87, and 0.94) exceeded those in Set_all (0.65, 0.73, and 0.6) by about 26, 19, and 57%, respectively.

**Table 5.** Performance measure indices of monocot species for the training and testing datasets by feature sets based on the same parameters.

| Type | Species | Feature | Precision | Accuracy | AUC |
|---|---|---|---|---|---|
| Monocots | *O. sativa* | Set_all | 0.78 | 0.83 | 0.9 |
| Monocots | *O. sativa* | Set1 | 0.76 | 0.81 | 0.9 |
| Monocots | *O. sativa* | Set2 | 0.76 | 0.81 | 0.88 |
| Monocots | *O. sativa* | Set3 | 0.76 | 0.81 | 0.93 |
| Monocots | *O. sativa* | Set4 | 0.76 | 0.81 | 0.9 |
| Monocots | *O. sativa* | Set5 | 0.79 | 0.83 | 0.9 |
| Monocots | *S. bicolor* | Set_all | 0.65 | 0.73 | 0.6 |
| Monocots | *S. bicolor* | Set1 | 0.65 | 0.73 | 0.6 |
| Monocots | *S. bicolor* | Set2 | 0.65 | 0.73 | 0.62 |
| Monocots | *S. bicolor* | Set3 | 0.65 | 0.73 | 0.62 |
| Monocots | *S. bicolor* | Set4 | 0.65 | 0.73 | 0.6 |
| Monocots | *S. bicolor* | Set5 | 0.82 | 0.87 | 0.94 |

Additionally, we further explored and compared these combined feature sets of four selected plant species, containing the eudicot and monocot species of evolutionary lineages. The results, plotted in Figure 5, strongly indicate that the featured protein of *pI*, which plays a vital role in determining molecular biochemical function, is essential for predicting OGs in eudicot genomes and further clarifying their biochemical function in eudicots via proteomic studies.

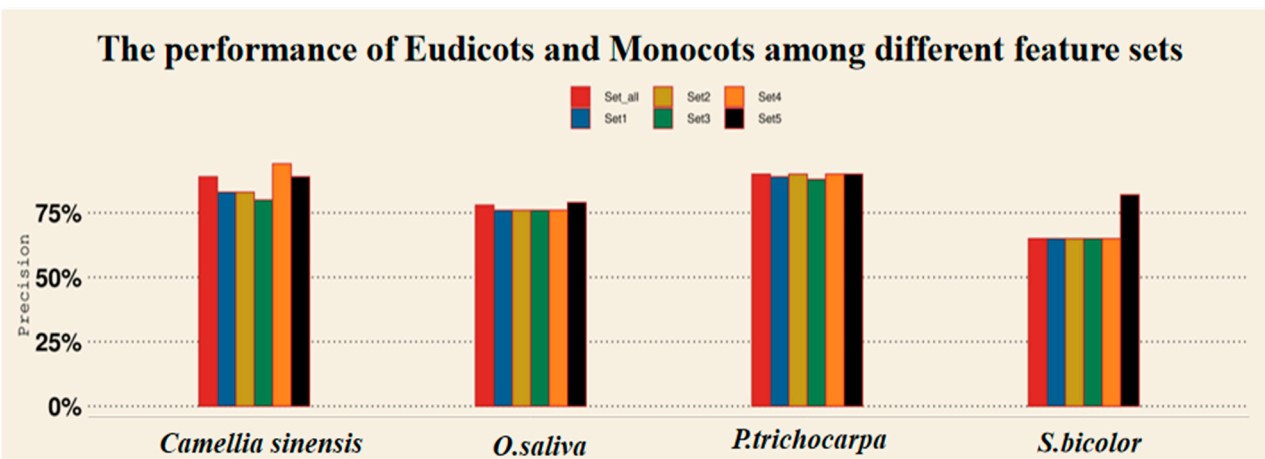

**Figure 5.** The performance precision for eudicots and monocots via different selected feature sets in angiosperm specie.

We also observed that GC content was more likely to impact prediction performance, as real OGs in monocot groups evolved from eudicots, such as *O. sativa* and *S. bicolour*. However, GC content is one of the critical compositional features of the genome and varies significantly among different genomes and regions within a genome [34,35].

Finally, to further validate the performance of the XGBoost-A2OG model for eudicot and monocot groups, we tested the model on the dataset of *Arabidopsis thaliana, Populus trichocarpa, Sorghum bicolour, Oryza sativa, Zea mays, Citrus sinensis,* and *Camellia sinensis* with feature set5 separately. The results are shown in Figure 6.

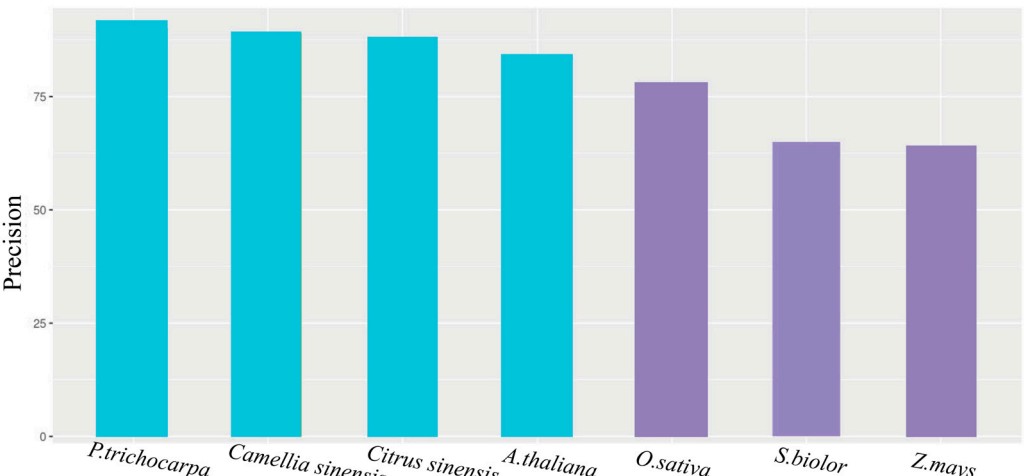

**Figure 6.** The testing performance in predicting OGs of various angiosperm species.

The precision of predicting OGs for different angiosperm species was not the same, indicating a higher reliability of XGBoost-A2OGs in identifying OGs of eudicot species (*P. trichocarpa*, *Camellia sinensis*, *Citrus sinensis*, and *A. thaliana*) than that of monocot species (*O. sativa*, *S. bicolour*, and *Z. mays*).

With a range of evolutionary processes, OGs can be derived in a lineage and provide lineage-specific adaptations. As mentioned above, there is some evidence that the sequence characteristics of orphan genes are common in two groups of angiosperm: eudicot and monocot species. However, some of them play different roles in identifying OGs based on the XGBoost-A2OG model due to differences in their evolution and origins. However, there is a lack of evidence on the mechanism of origin for the divergence of essential features of OGs between monocots and eudicots due to the rapid evolution of orphan genes.

## 5. Conclusions

Based on the background of enlarged genome sequences in angiosperm plants, this study proposed an XGBoost-A2OGs model to identify orphan genes (OGs) via the ensemble learning approach applied to several genome and cDNA features in angiosperm species, some of which have a consistent distribution. Cross-species models were trained on datasets of seven angiosperm species, performing better than SVM and other ensemble models (Adaboost, GBDT, and Random Forest). The proposed XGBoost-A2OGs method adopted makes multiple feature sets that have been proven helpful in OG identification and used feature selection to select the optimal feature subset. Thus, plant OGs exhibited discrepant results on combined features in eudicots (*P. trichocarpa* and *Camellia sinensis*) and monocots (*O. sativa* and *S. bicolour*) but still shared some features. Finally, the proposed method further established species-specific models with the optimal features on seven plants' datasets, which performed better on eudicot groups than on monocot ones.

In summary, XGBoost-A2OGs is a helpful method for identifying OGs from genome features. The feature importance of monocot and eudicot orphans was analyzed, providing a theoretical basis for the inheritance and variation of orphan genes in the process of evolution. In future work, with the rapid development of next-generation sequencing technologies, an ensemble learning approach with comparative genomics can be imported to obtain information on different types of angiosperm plants. Alternative deep learning algorithms, such as Transformer and LSTM, can also be applied to improve the potential performance. The follow-up study envisages incorporating some other essential features, such as gene expression, into the proposed model, which may significantly improve the efficiency of predicting OGs in angiosperm plants.

**Author Contributions:** Conceptualization, X.J.; methodology, Q.G.; software, X.Z.; H.Y.; writing—original draft preparation, Q.G.; writing—review and editing, Q.G. All authors have read and agreed to the published version of the manuscript.

**Funding:** This research was funded by commercial research fund named "High-throughput sequencing and metagenomic approaches for the study of functional health components of tea leaves" and grant number as 20223401002858.

**Data Availability Statement:** Not applicable.

**Conflicts of Interest:** The authors declare no conflict of interest.

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
