# Peer review of "Machine Learning-Based Prediction of Orphan Genes and Analysis of Different Hybrid Features of Monocot and Eudicot Plants"

_electronics, doi:10.3390/electronics12061433_

Round 1

Reviewer 1 Report

The article is devoted to the study of the classification of monocots and "true dicots" plants (eudicots) based on a group of morphological characters

The authors build a machine learning model in the article, which, based on the data, classifies the studied plants into 2 classes, using the XGBOOST algorithm and predicts the presence of orphan genes. The content decomposition corresponds to articles of this type: the authors select predictors, form datasets, implement the cross-validation process, and set the parameters of the machine learning model. Using ROC and AUC curves, the quality of the constructed model is assessed using various machine learning algorithms. This approach is standard and implemented by the authors in full. The obtained results testify to the effectiveness of the proposed classification and forecasting model.

Remark 1. The authors use open datasets about plants, but the article does not indicate the number of predictors and predictor data types. This is important because a number of machine learning algorithms work only with numerical data, and a number allow data to be measured in nominal scales (such as "small", "medium", "large").

Remark 2. It is not very clear what tools and libraries are used to implement this study (Python? R? SAP Hana?). Maybe attach code? This is important because the parameters of machine learning algorithms are the key characteristics of the built models. For example, the random forest algorithm does not have an OOB (out of bag) error. For the XGBOOST algorithm, the characteristics of the unique XGBOOST tree (branches, leaves, levels) and the value of the larning rate are important.

Remark 3. The article analyzes the results of 5 different algorithms. This is the correct approach. But why compare Ada Boost and XGBoost? In principle, it has been proven that the latter works better and classifies the data sample better. It's just overkill in my opinion.

Author Response

Dear Reviewer:

Thank you very much for your kindly comments on our manuscript (No. 2256149). Based on

reviewers’ suggestions, we carefully revised the manuscript.

We are now sending the revised article. Please see our point to point responses to all your

comments below, and the corresponding revisions in the body of manuscript, both marked

in red.

We hope the new manuscript will meet your standard.

Thank you again for your time and consideration.

Sincerely,

Qijuan Gao, Xiu Jin

Below, the original comments are in black, and our responses are in blue.

Remark 1. The authors use open datasets about plants, but the article does not indicate the number of predictors and predictor data types. This is important because a number of machine learning algorithms work only with numerical data, and a number allow data to be measured in nominal scales (such as "small", "medium", "large").

In this paper, the data type is discrete textual data, the applied data scales is medium.

Remark 2. It is not very clear what tools and libraries are used to implement this study (Python? R? SAP Hana?). Maybe attach code? This is important because the parameters of machine learning algorithms are the key characteristics of the built models. For example, the random forest algorithm does not have an OOB (out of bag) error. For the XGBOOST algorithm, the characteristics of the unique XGBOOST tree (branches, leaves, levels) and the value of the larning rate are important.

In this paper, the implement of this study is based on Python 3.9 which imported the tensorflow, sklearn package and so on. We have using the relevant parameter of machine learning algorithm to build this model.

Remark 3. The article analyzes the results of 5 different algorithms. This is the correct approach. But why compare Ada Boost and XGBoost? In principle, it has been proven that the latter works better and classifies the data sample better. It's just overkill in my opinion.

In this paper, adboost and xgboost are belong to ensemble learning method. There are some difference between the two method, for example, XGBoost is used decision tree, but the adboost is composed many week learner for learning. In particular, the xgboost have better applied in various area research, therefore, we have compared the result between this two method. Moreover, the comparison of the two methods is also intent to analyze their characteristic situations.

Reviewer 2 Report

In this paper, the authors proposed using XGBoost to identify OGs for seven angiosperm species based on hybrid features and XGBoost.

Several important issues are required to be addressed to improve the paper quality:

1-   In Section Abstract

-Splitting and dividing data makes the reader confused :

In line 17, the authors mentioned they used 5-fold cross-validation

In line 131, the author mentioned they used 7:3 ratio for training and testing datasets

In line 184, the author mentioned they used 10-fold cross-validation

2-   In Section 1. Introduction

-In lines 63-64, the authors mentioned that XGBoost is not yet used in predicting OGs. However, The same authors applied  XGBoost in predicting OGs in the following paper:

    Gao, Q., Jin, X., Xia, E., Wu, X., Gu, L., Yan, H., ... & Li, S. (2020). Identification of orphan genes in unbalanced datasets based on ensemble learning. Frontiers in Genetics, 11, 820.

This point needs clarification since applying XGBoost in predicting OGs is one of the main contributions of this article

3-    Section of related works is missing. I suggest adding separated Section related works  to discuss some existing works based on machine learning and how different the proposed method compared to these previous works

4-   By mistake, Figure 1 and Figure 2 have the same caption “Workflow of the framework”

5-   In Section 2.2. Ensemble algorithm, the authors only described general information about XGBoost : the loss function and some parameters of XGBoost. However, the important parts are missing such as the motivation and advantages of using XGBoost in predicting OGs, how XGBoost works, and how XGBoost is applied  in predicting OGs

6-   In Section 2.3. Data preparation and feature selection settings

-Is there any need for feature selection since there are only nine features?

-Why authors did not use any common filter feature selection techniques instead of manual feature selection?

7-   In Section 2.4. Validation strategies and evaluation metrics

It is important that authors write the equations of performance measures: accuracy, precision, recall, F1-Score, and AUC

8-   It is surprising that subsection 3.1 The model’s framework is put under Section 3 3. Results and Discussion while the explanation of steps of model in Section 2. Materials and Methods

I suggest moving the subsection 3.1 The model’s framework to Section 2. Materials and Methods

9-   In Section 4. Comparison with other methods for predicting cross-species OGs

In line 181, please describe the term “XGBoost-A2OGs” as it is the first time mentioned in the paper.

10-           In Section 5. Predicting OGs with different feature sets in eudicot and monocot 201 species via XGBoost-A2OGs

The same question. Why did the authors not use common filter feature selection techniques instead of manual feature selection?

11-   No results comparison with previous works. It is expected that authors compare the proposed method against previous works that used machine learning and feature selection methods used in the literature in predicting OGs.

Author Response

Dear Reviewer:

Thank you very much for your kindly comments on our manuscript (No.2256149). Based on

reviewers’ suggestions, we carefully revised the manuscript.

We are now sending the revised article. Please see our point to point responses to all your

comments below, and the corresponding revisions in the body of manuscript, both marked

in red.

We hope the new manuscript will meet your standard.

Thank you again for your time and consideration.

Sincerely,

QIjuan Gao, Xiu Jin

Thank you very much for your patience in reviewing the contents in paper.We have critically checked the paper for grammar and formatting issues. Below, the original comments are in black, and our responses are in blue.

1- In Section Abstract

-Splitting and dividing data makes the reader confused :

In line 17, the authors mentioned they used 5-fold cross-validation

In line 131, the author mentioned they used 7:3 ratio for training and testing datasets

In line 184, the author mentioned they used 10-fold cross-validation

We have already checked this article about the splitting and dividing data in lines 17,131 and 184 in the original article, and revised them on the updated version in lines 18, 205,226.

2- In Section 1. Introduction -In lines 63-64, the authors mentioned that XGBoost is not yet used in predicting OGs. However, The same authors applied XGBoost in predicting OGs in the following paper:

  Gao, Q., Jin, X., Xia, E., Wu, X., Gu, L., Yan, H., ... & Li, S. (2020). Identification of orphan genes in unbalanced datasets based on ensemble learning. Frontiers in Genetics, 11, 820.

This point needs clarification since applying XGBoost in predicting OGs is one of the main contributions of this article

We have re-written the expression to this section. In lines 66-69, the revision is “In addition, Gao et al have used effective SMOTE-ENN-XGBoost model to predict OGs of A. thaliana, However, to the best of the authors’ knowledge, it has yet to be carried out in the bioinformatic field of predicting OGs of different types of plant species.”

3- Section of related works is missing. I suggest adding separated Section related works to discuss some existing works based on machine learning and how different the proposed method compared to these previous works.

We have added the section of related works in lines 73-99, the revision is as following:

Recently, machine learning methods have received considerable interest in the identification of OGs fields, which are an important source of genetics and contribute to evolutionary innovations. These methods include a Decision tree (DT) [19], Neural network (NN) [19], Convolutional Neural Network(CNN) with transformer [20], and ensemble learning method [21]. Besides, many researchers have been conducted to compare different machine learning algorithms or combined with other methods to accelerate the performance of identification of OGs .

Gao et al proposed a novel ensemble method to predict OGs of A. thaliana in bioinformatics studies. Then another deep learning method, CNN with transformer technique was successfully applied to identifying OGs in moso bamboo which used a convolutional neural network in combination with a transform neural network in protein sequences [20]. Their proposed approach provides better performance in a specific species.

In addition, Decision tree and neural networks were employed to improve the accurate discovery of OGs by Casola et al relying on basic sequence features obtained from DNA and protein sequences in three angiosperm families. The experimental results showed that both DT and NN classifiers achieve high levels of accuracy and recall in identifying OGs.

Recently, many studies have confirmed that OGs generated de novo in a species may be more prevalent than gene duplication and be one of the main ways of orphan generation [22~26]. Some researchers have found that the newly evolved OGs in Arabidopsis, protein length is usually shorter, mainly due to the evolution of the orphan gene having fewer exons in the process, while in some species, the exon length is significantly shorter [27,28].

However, these researchers haven’t focused on different families of angiosperm plants. To find a general method to identify a large number of plants of OGs based on a rapid accumulation of genomic data, we have analyzed some features regarding the genome and protein sequences that may affect the results in the classification process.

.

  • By mistake, Figure 1 and Figure 2 have the same caption “Workflow of the framework

We have deleted the wrong captain in Figure 2. In line 118, the revision is “Fig 1: The phylogenetic tree of Monocotyledonous and eudicotyledonous plants”.

  • In Section 2.2. Ensemble algorithm, the authors only described general information about XGBoost : the loss function and some parameters of XGBoost. However, the important parts are missing such as the motivation and advantages of using XGBoost in predicting OGs, how XGBoost works, and how XGBoost is applied in predicting OGs.

We have added the detailed description about XGBoost, and how the XGBoost works in lines125-156.The revision is as following:

XGBoost (Extreme Gradient Boost) is an ensemble learning technique for regression and classification problems based on the boosting algorithm [18]. The motivation is to classify data using the best hyperplane representing the most extensive separation between two classes. Unlike the traditional integrated decision tree algorithm, XGBoost adds a regular term to the loss function, which can control the complexity of the model and prevent the model from overfitting. The objective function is given to be optimized by the following formula:

(1)Taylor's formula to approximate the original goal.

                               (1)

(2)Taylor expansion:

        (2)

(3)Among them, gi, hi are expressed as:

                                (3)

(4)The formula of decision tree complexity calculation:

                                (4)

(5)T is the number of leaf nodes, and w is the leaf node score. Substituting (2)-(4) into (1) the objective function:

               (5)

(6)Among,, which represents the sample set belonging to the j-th leaf node.

                   (6)

(7)In order to minimize the objective function, set the derivative being 0 and find the optimal prediction score of each leaf node:

                         (7)

(8) Substitute the objective function again to get its minimum value:

                 (8)

(9) Use obj to find the tree with the best structure and add it to the model, and apply the greedy algorithm to find the optimal tree structure. Each time when try to add a split to an existing leaf, the gain is calculated as follows:

       (9)

When the XGBoost model was actually used in the experiment, the following parameters were adjusted to make the model perform its best performance:

6- In Section 2.3. Data preparation and feature selection settings

-Is there any need for feature selection since there are only nine features?

-Why authors did not use any common filter feature selection techniques instead of manual feature selection?

We have added the common filter feature selection technique as supplementary method for comparing with the manual feature selection. In lines 179-195. The revision is as following:

In this paper, we divide into two parts feature selection, one is the filter-based feature selection. This algorithm adopts some principles involving information, consistency, dependency, and distance for measuring the feature characteristics, which are generalized for various classifiers based on the independent features of the machine learning algorithm [32]. For example, a variation filter is to remove the features with small difference value and retain the features with large variance value, because the variance of each feature determines the different degree of the feature in a sample. When a feature in the data set is exposed to Bernoulli distribution (binary classification), it can be used the formula as follow:

σ=p(1-p)                                           (10)

The classic Chi-square(Chi2) filter method is a statistical test for computing the correlation from two types of categorical data. Considering the inconsistency between the observed value and the expected value of the sampling frequency, such as the independent variable equal to i and the dependent variable equal to j, the statistic is constructed, Chi2 tests use the following formula to calculate the test statistic:

                                        (11)

7- In Section 2.4. Validation strategies and evaluation metrics

It is important that authors write the equations of performance measures: accuracy, precision, recall, F1-Score, and AUC

We have added the equations of performance measures: accuracy, precision, recall, F1-Score, and AUC. In lines 219-242.The revision is as following:

The confusion matrix is a matrix table (shown in Table 1) that is used to judge the validation of classification. The results of the prediction model are analyzed using four basic indicators: true positive (TP), true negative (TN), false positive (FP) and false negative (FN).

Real Positive

Real Negative

Predict positive

TP

FP

Predict negative

FN

TN

We performed an initial statistical analysis to evaluate the prediction performance for binary classes and grasp the critical features. As the performance measures, stratified five-fold cross-validation was used for obtaining classification accuracy; however, accuracy was found to be an inappropriate evaluation metric for class-imbalanced datasets. Alternatively, precision, recall, F1-Score, and AUC (area under the ROC curve) parameters were used to evaluate the proposed method’s feasibility, as in [33]. The AUC is the value of the area under the ROC curve (receiver operating characteristic) that reflects the probability of identifying correct and wrong results according to different thresholds, which is generally between 0.5-1. The quantized index value can better compare the performance of the classifiers: a high-performance classifier AUC value is close to 1properly reflected the test performance.

(i)Accuracy rate (accuracy rate of positive samples):                                 (12)

(ii)Recall rate (accuracy rate of positive samples):

                                  (13)

(iii)Precision (precision rate of positive samples):

                                  (14)

(iv)F1-score value:

F1                                     (15)

8- It is surprising that subsection 3.1 The model’s framework is put under Section 3 3. Results and Discussion while the explanation of steps of model in Section 2. Materials and Methods

I suggest moving the subsection 3.1 The model’s framework to Section 2. Materials and Methods

Thank you for your good suggestion, and we have moved this part of framework to Result and discussion, In line 102-106, shown in red part subsection 3.1.

9- In Section 4. Comparison with other methods for predicting cross-species OGs

In line 181, please describe the term “XGBoost-A2OGs” as it is the first time mentioned in the paper.

The term “XGBoost-A2OGs” have revised in section of abstract in line 15-16. The revision is “This paper presents an automated predictor, XGBoost-A2OGs (identification of OGs for angiosperm based on XGBoost) ”

10-In Section 5. Predicting OGs with different feature sets in eudicot and monocot 201 species via XGBoost-A2OGs

The same question. Why did the authors not use common filter feature selection techniques instead of manual feature selection?

We have used the common filter feature selection techniques as a supplementary test for this study, and the result have been added in line 289-312, shown as the following:

Some features might become noise, deteriorating the robustness and stability of the constructed model. Moreover, contribution rates of various features differ, the highest ones being the most lucrative for OGs’ prediction. Therefore, this work presents two filter-based selection methods to remove irrelevant and redundant features in terms of both training processes. In particular, we selected two types of delegated species from the eudicot subclass (P. trichocarpa and Camellia sinensis) and monocot subclass (O. sativa and S. bicolor) applied with filter-based selection methods. Then the filtered feature are the same containing the GC, protein length, Mw(Da), pI. Thus, the classification results on these selection methods with four species separately by variation and Chi2 method based on XGBoost-A2OGs model are listed in Table 3:

Table 3. Performance measure indices of eudicot and monocot species for the training and testing datasets by filter method based on the same parameters.

Type

Species

Filter method

Precision

Accuracy

AUC

Eudicots

P.trichocarpa

 variation

0.92

0.93

0.94

Eudicots

P.trichocarpa

Chi2

0.9

0.92

0.94

Eudicots

Camellia sinensis

 variation

0.82

0.69

0.85

Eudicots

Camellia sinensis

Chi2

0.82

0.69

0.85

Monocots

O.sativa

 variation

0.78

0.83

0.9

Monocots

O.sativa

Chi2

0.78

0.83

0.9

Monocots

S.bicolor

 variation

0.81

0.87

0.94

Monocots

S.bicolor

Chi2

0.81

0.87

0.94

Filter algorithm has ability to scale for multiple dimensional datasets. However, the features selected by the filter method ignores the interaction among features and individual scores in filter-based method are assigned to each feature without considering its significance in combination with other shared features. Therefore, we further proposed artificial group for feature selection to explore the contribution of each feature for different types of angiosperm. First of all, we also selected eudicot subclass (P. trichocarpa and Camellia sinensis) and applied to them five sets of feature selection methods in order to identify the one with the optimal performance. The classification results on five sets of feature selection methods with two species separately based on XGBoost-A2OGs are listed in Table 2, where the Set3 of Camellia sinensis featured the lowest precision, accuracy, and AUC values (0.80, 0.69, and 0.85). Meanwhile, the Set5 of P.trichocarpa combined the highest respective values (precision of 0.9, accuracy of 0.92, and AUC=0.94)

  • No results comparison with previous works. It is expected that authors compare the proposed method against previous works that used machine learning and feature selection methods used in the literature in predicting OGs.

This paper is a discussion of the features of monocotyledons and dicotyledons in the previous research. The core of its focus is mainly on the differentiation analysis of the discriminative features of the different species. In particular, monocots diverged from their eudicot relatives. Therefore, we compared the gene with protein features of the two families, and explore the feature importance analysis of the two plant types based on the previous works.

Reviewer 3 Report

The study addresses an interesting topic, is well written with an introduction that describes the evolution of knowledge at the time, well-defined objectives, an adequate methodological approach and an interesting presentation and discussion of the results. Conclusions are based on the results. That is why we are in favor of its publication in this prestigious periodical.

Author Response

Dear Reviewer:

Thank you very much for your kindly comments on our manuscript (No. 2256149). Based on

reviewers’ suggestions, we carefully revised the manuscript.

We are now sending the revised article. Please see our point to point responses to all your

comments below, and the corresponding revisions in the body of manuscript, both marked

in red.

We hope the new manuscript will meet your standard.

Thank you again for your time and consideration.

Sincerely,

Qijuan Gao, Xiu Jin

Reviewer 4 Report

In the manuscript, the authors utilized different models to identify orphan genes. The study is well-designed and the performance of the XGBoost model is compared with other models for cross-validation. The study is important as mentioned that orphan gene identification is time-consuming. The manuscript suggested a machine-learning prediction of orphan genes and the method can be further improved with other features for example gene expression data etc. Overall, the authors tested the models and validated the conclusion with multiple evaluation metrics to support their findings. The manuscript is suggested to be accepted.  

Author Response

(The authors gave the same response as above.)

Round 2

Reviewer 2 Report

The authors addressed the comments in a good way. I found two simple mistakes in the revised article:

-In line 63, the authors should cite Gao et al.

- In lines 156-159,  mistakes in English in the following sentence:

“ When the XGBoost model was actually used in the experiment, the following parameters were adjusted to make the model perform its best performance: Several parameters were adjusted when using the XGBoost model in an actual experiment to optimize the model performance”

Author Response

Dear Reviewer:

Thank you very much for your kindly comments on our manuscript (No. 2256149). Based on

reviewers’ suggestions, we carefully revised the manuscript.

We are now sending the revised article. Please see our point to point responses to all your

comments below, and the corresponding revisions in the body of manuscript, both marked

in red.

We hope the new manuscript will meet your standard.

Thank you again for your time and consideration.

Sincerely,

Qijuan Gao, Xiu Jin

Below, the original comments are in black, and our responses are in blue.

-In line 63, the authors should cite Gao et al.

In this paper, I have added the cite information in line 63.

- In lines 156-159, mistakes in English in the following sentence:

“ When the XGBoost model was actually used in the experiment, the following parameters were adjusted to make the model perform its best performance: Several parameters were adjusted when using the XGBoost model in an actual experiment to optimize the model performance”

The following sentence has wrote repeated, therefore, I have deleted the last sentence and re-wrote it as “When the XGBoost model was actually used in the experiment, the following parameters were adjusted to make the model perform its best performance.” In lines 156-157.
